# Assessing Community Readiness to Use Social Platforms for Stroke Survivors’ Recovery and Caregiver Support in Saudi Arabia

**DOI:** 10.3390/nursrep15090325

**Published:** 2025-09-08

**Authors:** Lisa A. Babkair, Mohammed Al-Sahabi, Husam Al-Ghamedi, Abdulmajeed S. Al-shehri, Ahmed Al-Zeer, Faygah Shibily, Rwan M. Alyafee

**Affiliations:** Faculty of Nursing, King Abdulaziz University, Jeddah 21589, Saudi Arabia; malsahabi0002@stu.kau.edu.sa (M.A.-S.); hahmedalghamedi@stu.kau.edu.sa (H.A.-G.); aalshehri1536@stu.kau.edu.sa (A.S.A.-s.); aalzeer0008@stu.kau.edu.sa (A.A.-Z.); fshibily@kau.edu.sa (F.S.); ralyafee@kau.edu.sa (R.M.A.)

**Keywords:** stroke, stroke survivors, caregivers, social platforms, support

## Abstract

**Background:** Stroke survivors and their family caregivers face substantial burdens resulting from a complex interplay of clinical, functional, and psychosocial factors. Community-based resources and social networking are critical for alleviating caregiver burden and improving outcomes for both caregivers and stroke survivors. **Objectives:** To assess the Saudi population’s readiness to use social platforms for stroke survivors’ recovery and caregiver support. **Methods:** A cross-sectional descriptive design was employed between March and June 2025 to collect data from community-dwelling individuals using sociodemographic and Technology Readiness Index (TRI) questionnaires. **Results:** A convenience sample of 576 participants was enrolled in this study. Overall, the participants showed a high level of technology readiness, with a total TRI mean score of M = 3.65, SD = 0.86. Optimism had the highest mean score, followed by innovativeness, insecurity, and discomfort. Significant differences in technology readiness were identified based on age, educational level, employment status, familiarity with modern technology, and healthcare provider status. **Conclusions:** This study demonstrates a high level of technological readiness across the Saudi population, indicating strong potential for integrating social connection platforms into stroke recovery and caregiver support. These findings align with Saudi Arabia’s Vision 2030 goals for digital transformation in the healthcare sector. Future research should focus on conducting feasibility studies to better understand the desirable features of e-health services and digital solutions within the Saudi community.

## 1. Introduction

Stroke is the second leading cause of death and the third leading cause of death and disability combined, accounting for over 160 million disability-adjusted life years (DALYs) lost each year worldwide [1]. Despite advances in acute stroke care, many survivors experience persistent physical, cognitive, and psychosocial impairments. These disabilities place a significant burden not only on individuals and families but also on healthcare systems, particularly in low- and middle-income countries where rehabilitation resources are limited [1].

Stroke rehabilitation services are essential for recovery, improving quality of life, and reducing the societal and economic impacts of stroke [2]. Globally, nearly 94 million people have survived a stroke [2]. The increasing prevalence of stroke worldwide highlights the urgent need to expand access to multidisciplinary rehabilitation services and community-based support models extending beyond hospital discharge.

In Saudi Arabia, the annual incidence of stroke ranges from 19 to 57 cases per 100,000 individuals, with variations attributed to differences in access to healthcare and reporting across regions [3]. The prevalence of stroke has been rising, reflecting increases in risk factors such as diabetes, hypertension, and population aging [4]. The one-year mortality rate after stroke in Saudi Arabia is between 14% and 27%. The incidence of stroke is projected to increase by 60% over the next decade, further highlighting the urgent need for enhanced rehabilitation and preventive strategies [4].

Stroke survivors and their family caregivers face substantial burdens resulting from the complex interplay of clinical, functional, and psychosocial factors [5]. Stroke survivors often experience significant neurological impairment, higher levels of dependence in daily activities, cognitive decline, and comorbidities such as hypertension and diabetes. These outcomes are a heavy burden for survivors and caregivers [6]. In turn, family caregivers face greater burdens and challenges when providing care to survivors with severe disability or cognitive impairment and when caregiving extends over months or years [7,8]. Socioeconomic factors, including lower household income and limited access to resources, further exacerbate the strain on both survivors and caregivers [7,8]. However, social and community support—such as financial aid, help from family, and access to rehabilitation services—has been shown to alleviate burden, improve satisfaction, and enhance overall well-being for both stroke survivors and their caregivers [6,7]. Community-based resources and social networking play a critical role in reducing caregiver burden and improving outcomes for both caregivers and stroke survivors, offering emotional support, practical advice, and a sense of connection which can reduce feelings of isolation and foster engagement for both groups. The use of telemedicine and online platforms increases access to ongoing support and rehabilitation, especially for those in rural or underserved areas [8]. These platforms enable survivors and caregivers to share experiences, access information, and receive emotional support from a larger network, helping to alleviate burdens associated with stroke recovery [9]. However, the roles and effectiveness of these platforms in providing continuous support for stroke survivors and their family members remain under investigation [10].

Few researchers have investigated the roles of online community platforms in supporting stroke survivors and their families. The American Stroke Association has developed an online platform for social connection and support targeting this population. This platform allows users to search for both in-person and virtual support groups, connect with others who have experienced similar challenges, share experiences, and receive emotional support and practical advice [11]. Similarly, the United Kingdom Stroke Association offers virtual support groups, weekly volunteer calls, online activities, and a confidential helpline. These online groups create safe spaces for survivors and their families to connect, share, and rebuild confidence after stroke [12]. Despite their potential advantages, many communities remain unprepared to adopt these tools effectively. The use of online social connection platforms is affected by factors including digital literacy, time since stroke, emotional readiness, and the availability of caregiver support. In one study, approximately 86% of stroke survivors and caregivers had internet access, with caregivers more likely to use the internet than survivors [13]. Younger age and longer time since stroke (>12 months) were associated with higher rates of internet access, and smartphones were most commonly used. Future feasibility and acceptability studies should consider factors influencing community readiness when designing and implementing web-based interventions for stroke survivors [13].

The effectiveness of these digital platforms relies heavily on the community’s readiness to adopt and engage with them. Factors such as platform usability and accessibility, attitudes toward online health support, and content clarity can significantly influence participation. For instance, older caregivers and stroke survivors may struggle with technology, limiting their ability to benefit from these resources [13]. These barriers highlight the need to assess community readiness to adopt digital health tools in order to maximize their potential.

Despite the growing number of stroke survivors, structured rehabilitation and support resources remain limited for survivors and their caregivers in Saudi Arabia. Digital engagement now spans all age groups, making online platforms accessible and inclusive tools for health-related support. Saudi Arabia’s digital transformation, driven by Vision 2030 and substantial government investment, is fundamentally reshaping healthcare by integrating advanced technologies, such as telemedicine, artificial intelligence, and electronic health records [14]. This transformation is making healthcare more accessible, efficient, and patient-centered, particularly benefiting those in remote or underserved regions. The rapid expansion of high-speed internet, mobile health applications, and digital health platforms, such as the Sehhaty app, demonstrates the Kingdom’s commitment to bridging gaps in care and empowering citizens to manage their health from home. Within this context, nurses are pivotal in leveraging digital platforms to extend patient education, provide psychosocial support, coordinate rehabilitation care, and empower caregivers. Therefore, community readiness to use social connection platforms to support stroke survivors and their caregivers is essential for guiding nursing interventions and shaping evidence-based practice.

This study aims to help in determining the preparedness of individuals and communities to engage with digital health solutions. Such assessments provide valuable insights into digital literacy, access to technology, motivation, and trust, all of which influence the successful adoption and sustained use of online platforms. Through identifying strengths and barriers at the community level, such studies enable the design of tailored interventions that match actual needs and readiness, ensuring efficient resource use and increasing the likelihood of success. Without an understanding of readiness, there is a risk of implementing digital solutions that are underutilized or ineffective, potentially wasting resources and missing opportunities to improve stroke recovery and caregiver support. The aim of this study is to fill the gap in the literature by assessing the readiness of the Saudi community to utilize social platforms for stroke survivors’ recovery and caregiver support. We hypothesize that most individuals in the Saudi community would demonstrate a positive technology readiness profile, suggesting that they are prepared to use social platforms to support stroke survivors and their caregivers.

## 2. Materials and Methods

### 2.1. Study Design

In this study, we employed a quantitative, descriptive, cross-sectional design to collect data on community readiness to utilize social connection platforms for stroke survivors and caregiver support. Participants’ sociodemographic characteristics were examined to determine whether they influenced TRI levels.

### 2.2. Settings and Sample

Data were collected using an online survey distributed via the social media platform WhatsApp. A convenience sample of 576 community members aged 18 years and older who were citizens or residents of Saudi Arabia participated in the study. The study focused on members of the general public to obtain insights into community readiness to use social connection platforms to support stroke survivors. The response rate was 78%, and 18 people who did not complete the survey were excluded from the analysis. A large sample size allows for representation of the target population, ensuring that the findings are generalizable to the community. This sample size is adequate for public health or community engagement research, as it provides sufficient diversity and statistical power to detect meaningful differences across subgroups or population segments.

### 2.3. Recruitment and Data Collection

Participants were recruited after obtaining ethical approval for the study. The principal investigator distributed the study information and the Survey Monkey link to the public via WhatsApp. The researchers monitored data collection between March and June of 2025 until the target sample size was achieved. The data were collected using two questionnaires.

### 2.4. Instruments

The sociodemographic questionnaire included variables such as age, gender, nationality, marital status, literacy level, monthly income, and employment status.

The second questionnaire focused on nursing practices with stroke patients and was developed using the Technology Readiness Index (TRI). The TRI is a psychometric scale developed to assess an individual’s propensity to embrace and use new technologies in various aspects of life, including home and work environments. It reflects a person’s overall mindset, shaped by mental enablers and inhibitors that influence their willingness to adopt technologies. The TRI was initially introduced by A. Parasuraman in 2000 and later updated to the streamlined TRI 2.0 in 2015 [15,16], which contains 16 attributes designed to measure technology readiness more efficiently. The TRI consists of four key dimensions that capture willingness to adopt new technologies. Optimism reflects a positive outlook, highlighting beliefs that technology enhances control, flexibility, and efficiency. Innovativeness represents a tendency to be a technology pioneer, eager to experiment with and lead the adoption of new tools. Discomfort reflects feelings of being overwhelmed or lacking control over technology, while insecurity reflects distrust of and skepticism toward the reliability and security of technological solutions. These dimensions provide a comprehensive view of the psychological factors influencing technology adoption, acknowledging that individuals can simultaneously hold both positive and negative beliefs about technology.

The Arabic version of the TRI was used in this study. The internal consistency of the TRI was evaluated in this study using Cronbach’s alpha. The overall scale demonstrated excellent reliability with a Cronbach’s alpha of (α = 0.92). Reliability analyses for the four subscales also indicated acceptable to good internal consistency: optimism (α = 0.83), innovativeness (α = 0.70), discomfort (α = 0.77), and insecurity (α = 0.71). These results suggest that the TRI and its subscales are reliable tools for assessing technology readiness in the current study population. The psychometric properties of the TRI have been examined and show that the scale is a reliable and valid tool for assessing individuals’ readiness to adopt new technologies [15,16]. TRI 2.0 demonstrates good reliability (α = 0.70–0.83) and sound validity, with strong evidence for its four-dimensional structure and predictive power for technology-related behaviors. The reported Cronbach’s alpha values indicate good internal consistency across the four dimensions of the TRI: optimism (α = 0.80), innovativeness (α = 0.83), discomfort (α = 0.70), and insecurity (α = 0.71) [16]. The total score is categorized as low (TRI ≤ 2.89), medium (TRI between 2.90 and 3.51), and high (TRI > 3.51) based on aggregate scores. The same cut-off point was applied to the subscales and the global TRI score, as all items use the same 5-point response scale and no separate, validated cut-off exists for each subscale in the original instrument [15,16].

### 2.5. Data Analysis

To determine an adequate sample size for the study, an a priori power analysis was conducted using G*Power version 3.1. The analysis indicated that a sample size of N = 200 was required to achieve 80% power to detect a medium effect at a significance level of α < 0.05 using one way ANOVA. Data were analyzed using SPSS version 26, with descriptive statistics used to explain the sample and the study instruments. Means, standard deviations, and frequencies were calculated for demographic variables and TRI scores. Bivariate analyses were conducted to identify factors associated with technology readiness level.

## 3. Results

### Sociodemographic Characteristics

A sample of 576 participants completed the online questionnaire. Table 1 presents the demographic characteristics of the sample. Participants were middle-aged overall (M = 40.90, SD = 12.77), with ages ranging from 20 to 76 years. The majority of participants (333, 57.8%) were male and held Saudi nationality (432, 75%). Marital status was almost equally distributed between married (260, 45.1%) and single (286, 49.7%) participants. Most participants (401, 69.6%) were university graduates, and 275 (47.7%) were employed. Most rated their familiarity with using technology as very good (234, 40.6%). Participants were distributed across Saudi Arabia, with most residing in the Makkah region (216, 37.5%).

With regard to health-related conditions, 118 (20.5%) reported having survived a stroke and 227 (39.4%) reported the presence of a friend or relative who had had a stroke. In terms of healthcare specialty, 130 (22.7%) identified as a healthcare provider, while 69 (12%) were nurses.

Table 2 presents participants’ overall level of technological preparedness, which was high, with a total TRI mean score of M = 3.65 (SD = 0.86). Optimism had the highest mean score at M = 3.71 (SD = 1.11), indicating that participants generally believe in the benefits of technology and expect it to improve their lives. Innovativeness followed closely, with a mean of M = 3.70 (SD = 0.83), showing that many participants consider themselves early adopters or are enthusiastic about new technologies. Insecurity and discomfort, typically considered barriers to technology adoption, had mean scores of M = 3.63 (SD = 0.87) and M = 3.57 (SD = 0.96), respectively. Although these scores are in the medium range, they indicate that some users experience hesitation, fear of technical complexity, or concerns about reliability and trust.

Further statistical analysis was conducted to determine whether demographic or health-related characteristics significantly impacted participants’ TRI scores. The results showed no statistically significant differences in TRI scores according to nationality, gender, marital status, stroke survivability, or having a stroke-affected relative, indicating that these factors had no bearing on technological readiness. However, younger participants demonstrated greater technological readiness than older participants, indicated by a significant negative relationship between age and TRI scores (r = −0.29, *p* = < 0.001; see Table 3).

One-way analysis of variance (ANOVA) and post hoc testing revealed a statistically significant difference in TRI scores based on educational level, F(3, 572) = 397.78, *p* < 0.001, with participants holding a university degree or higher demonstrating significantly higher TRI scores than those with a lower level of education. Similarly, there was a significant difference in TRI scores based on employment status, F(4, 571) = 14.10, *p* < 0.001, with employed participants reporting greater readiness than other groups. Additionally, familiarity with modern technology was significantly associated with TRI scores, F(4, 571) = 95.58, *p* < 0.001. Participants who rated their technology skills as excellent or very good scored higher on the TRI compared to other groups. Finally, an independent samples t-test showed that healthcare providers (M = 3.83, SD = 0.68) had significantly higher TRI scores than non-healthcare providers (M = 3.62, SD = 0.90), t(274) = [2.93], *p* < 0.05; see Table 4.

## 4. Discussion

The purpose of this study was to assess community readiness to utilize social platforms for stroke survivors’ recovery and caregiver support in Saudi Arabia. The findings revealed a high level of technology readiness, reflecting a strong propensity among participants to embrace and use new technologies in their daily lives. This result is consistent with the study of Ortiz-Fernandez et al. (2021), who examined the attitudes and experiences of chronic stroke survivors in the community regarding the use of technology to support their health and rehabilitation at home [17]. We found that most stroke survivors expressed a willingness to use technology as a tool to improve their health status, with the majority of participants eager to use technology for self-management, information gathering, and rehabilitation exercises, though preferences varied by age, education, and disability level. Although we did not use the TRI instrument explicitly, we directly assessed technology readiness and adoption attitudes among stroke patients living in the community, providing strong evidence that stroke survivors are open to and capable of engaging with technological interventions to support their recovery and daily living [17]. A previous study has also evaluated the use of a virtual community of practice to support regional stroke care best practice implementation in an urban context [18]. The researchers reported that the virtual community provided immediate value in supporting user networking, community activities, and interactions. Furthermore, a systematic review and meta-analysis reported that technology-based psychosocial interventions are effective in enhancing self-efficacy and caregiving competence, as well as alleviating anxiety and depression among family caregivers of stroke survivors [10]. The positive impacts of technology, such as social connection platforms, on stroke survivors’ outcomes and caregiving support after hospital discharge highlight the urgent need to implement these technologies, especially given the high community readiness scores for technology use found in this study.

In this study, participants demonstrated particularly high scores in the dimensions of optimism and innovativeness, reflecting favorable perceptions of technology and a willingness to embrace novel digital tools. These results are consistent with those of Zhang et al. (2023) [9], who investigated the factors influencing patients’ adoption of online health communities from a patient-centered perspective. The authors found that perceived usefulness, ease of use, social support, trust, and health-related motivation significantly predicted patients’ intention to engage with online health communities [9]. Lemke et al. (2020) also found that people with stroke are often highly motivated to use information and communication technology devices in their daily lives [19].

Although insecurity and discomfort are viewed as barriers to technology adoption, in our sample, these were in the medium range, suggesting that participants did not strongly perceive them as barriers. The mean score for insecurity reflects concerns about trust, data privacy, or misuse when using technology. A mean of 3.63 on a 5-point scale indicates a medium—but not extreme—level of concern. This implies that although users may have some doubts, these are not severe enough to prevent the adoption of new technology, especially when compared to their high optimism and innovativeness scores. Regarding the mean score for discomfort, this indicates feelings of being overwhelmed or lacking control over new technologies. A medium score suggests that some users may feel uncomfortable but, on average, their discomfort is not high enough to significantly hinder readiness. These findings are consistent with a study that investigated problems and barriers related to the use of mobile health apps [20]. Use of digital health technologies raises concerns regarding data security and privacy risks, and there is a lack of transparent communication concerning data privacy policies, which are often unclear or missing entirely. Patients and healthcare providers often express significant concerns regarding data privacy and security, disclosing personal data, unauthorized access, breaches of confidentiality, and fears that stored data could be misused or sold by providers [21]. Healthcare workers also highlight the risk of unauthorized identification, and patients fear the unpredictable effects of data leaks [22].

In this study, we found a significant relationship between technology readiness levels and participants’ age, education level, employment status, familiarity with modern technology, and healthcare provider status. Specifically, younger participants demonstrated higher levels of technology readiness compared to older participants. These findings align with a study by Kuo et al. (2025), who reported that older individuals often experience greater discomfort and insecurity when using digital health technologies [23]. Likewise, Ortiz-Fernandez et al. (2021) found that 100% of younger patients  ≤ 45 years old showed greater readiness to install digital devices at home compared to 52% of participants ≥ 65 years old, a statistically significant difference (*p* = 0.025) [17]. Stroke-related and age-associated impairments can limit survivors’ ability to use and benefit from these technologies [19].

Educational level also plays a crucial role in adopting new technology. We found that participants with a university degree or higher showed greater technology readiness compared to those with a lower level of education. Our findings are consistent with a study that assessed attitudes toward the use of technology to support chronic stroke survivors in a home-based setting [17]. Ortiz-Fernandez et al. (2021) [17] found that participants with higher levels of education, such as university graduates, had a greater tendency to install digital devices and share information compared to those with no education.

Employed participants showed greater readiness to use social platforms compared to other groups, and healthcare providers showed a higher technology readiness level compared to non-healthcare providers. Our findings are strongly supported by Saudi Arabia’s 2030 Vision, in which remarkable change and growth are driven by the National Transformation Program [24]. Furthermore, the Healthcare Sector Transformation Program, driven by Vision 2030, aims to improve access to health care, expand e-health services and digital solutions, improve quality of care, and adhere to international standards [25]. Another study has investigated healthcare providers’ perceptions and barriers to use of telehealth applications in Saudi Arabia, reporting that telehealth was perceived as positive as well as valuable and confidential for monitoring and providing care [26]. Due to the force of healthcare digital transformation in Saudi Arabia, the majority of individuals are utilizing technology in their daily lives. Virtual clinical visits to provide regular follow-ups for people with chronic diseases increased after the COVID-19 pandemic compounded the community’s motivation to learn about using smartphones. Furthermore, most governmental services in Saudi Arbia transitioned to digital formats via apps with virtual customer support, which explains our findings regarding individual familiarity with modern technology [24].

### 4.1. Limitations of This Study

This study has some limitations. First, because we used a cross-sectional design, we were unable to evaluate how participants’ technological preparedness might change over time. Second, convenience sampling may introduce selection bias and reduce the sample’s representativeness, limiting the results’ applicability to the larger Saudi population. Third, online surveys distributed via social media may have skewed the sample toward respondents who were more tech-savvy by excluding people with worse digital literacy, less internet access, or lower use of technology. Additionally, the number of stroke survivors and family caregivers in this sample was small, which may affect the generalizability of the findings regarding the stroke population’s readiness to use social platforms. Moreover, healthcare providers represented a larger proportion of the sample than their actual distribution within the Saudi Arabian population, which may affect the generalizability of the findings. However, their inclusion also enriched the results by providing perspectives from both professionals and the general community. Additionally, the response rate from some regions of Saudi Arabia was relatively small, which may limit the ability to capture regional differences in community readiness. However, the study’s large sample size and use of validated tools, despite these methodological limitations, improve the findings’ internal validity and offer valuable insights into the community’s readiness for digital health interventions aimed at stroke recovery and carer support.

### 4.2. Implications and Recommendations for Nursing Practice

The participants’ TRI scores were high, indicating that the Saudi community is ready to establish a social connection platform for stroke survivors’ recovery and caregiver support. Online social connection platforms offer significant benefits for stroke survivors and their caregivers during the recovery process [8,9]. By facilitating communication and interaction with peers, health care professionals, and support groups, these platforms help to reduce feelings of isolation and provide much-needed emotional support. Stroke survivors can share their experiences, receive encouragement, and access valuable information about rehabilitation and self-care, which can enhance motivation and improve recovery outcomes. For caregivers, online communities provide a space to exchange practical advice, seek guidance, and connect with others who understand their challenges, reducing stress and increasing their confidence in providing care. Overall, the use of online social platforms fosters a sense of community and belonging, which is essential for emotional well-being and successful rehabilitation after a stroke [8,9]. Online social platforms can take several forms—for example, the Ministry of Health could create a dedicated page on its website to support stroke survivors and their caregivers, or a purpose-built app could be developed to facilitate peer connection and information sharing.

Nurses and healthcare providers play a vital role in supporting online social connection platforms for stroke survivors and their caregivers across several dimensions, including education, clinical practice, policymaking, and future research. In education, they empower patients and caregivers by providing accessible health information, teaching self-management skills, and promoting health literacy through digital resources. In clinical practice, they use these platforms to coordinate care, monitor recovery, support adherence to treatment plans, and facilitate virtual support groups, all of which contribute to better rehabilitation outcomes and reduced caregiver burden. Nurse-led community education programs may be effective in enhancing familiarity and confidence with social platforms among older adults or those with limited digital experience. These strategies aim to reduce disparities in readiness and ensure that social connection platforms can be inclusive and beneficial for all community members, particularly stroke survivors and their caregivers., In policymaking, nurses advocate for the integration of online support into standard stroke care, help establish protocols to ensure quality and privacy, and contribute to the development of ethical guidelines for digital health. Future research should include feasibility studies to gain insight into the desirable features of e-health services and digital solutions. They should also focus on evaluating the effectiveness and visibility of online social platforms and addressing the barriers to utilizing these platforms among the Saudi community.

## 5. Conclusions

This study demonstrated a high level of technological readiness among the Saudi population, indicating a promising foundation for incorporating online social connection platforms into stroke recovery and caregiver support in line with Saudi Arabia’s Vision 2030 for digital transformation of the healthcare sector. The nation’s rapid progress in digital infrastructure and e-health initiatives suggests that online platforms can be effectively integrated into healthcare strategies to enhance rehabilitation outcomes and foster stroke survivors’ recovery. Future research should focus on evaluating the effectiveness and visibility of online social platforms and addressing barriers to utilizing the platforms among the Saudi community.

## Figures and Tables

**Table 1 nursrep-15-00325-t001:** Sociodemographic characteristics of the sample.

Characteristics	N	Minimum	Maximum	Median	Mean	Std. Deviation
Age	576	20	76	43	40.90	12.774
Characteristics					Caregivers (N = 576)	
					N	(%)
Sex			Male		333	57.8
			Female		243	42.2
Nationality			Saudi		432	75
			Non-Saudi		144	4.0
Marital Status			Married		260	45.1
			Single		286	49.7
			Divorced		22	3.8
			Widow		8	1.4
Education			Middle School		11	1.9
			High School		136	23.6
			University		401	69.6
			Higher Degree		28	4.9
Employment			Employed		275	47.7
			Unemployed		55	9.5
			Retired		45	7.8
			Housewife		69	12.0
			Student		132	22.9
Self-Rating of Familiarity With Using Technology			Poor		2	0.3
			Fair		18	3.1
			Good		126	21.9
			Very Good		234	40.6
			Excellent		196	34.0
Residence in Saudi Arabia			Medina		170	29.5
			Makkah		216	37.5
			Riyadh		62	10.8
			Eastern Saudi Arabia		75	13.0
			Asir		22	3.8
			Jazan		8	1.4
			Tabuk		7	1.2
			Al-Baha		7	1.2
			Najran		4	0.7
			Al-Jawf		3	0.5
			Hail		2	0.3
Stroke Survivor			Yes		118	20.5
			No		458	79.5
Relative or Friend of Stroke Survivor			Yes		227	39.4
			No		349	60.6
Healthcare Provider			Yes		130	22.6
			No		446	77.4
Healthcare Provider Specialty			Nurse		69	12.0
			Doctor		23	4.0
			Respiratory Therapist		21	3.6
			Physiotherapist		17	3.0
			Other		446	77.4

Note: N = number of participants.

**Table 2 nursrep-15-00325-t002:** Average score for TRI and subcategories.

	N	Minimum	Maximum	Mean	Std. Deviation
Optimism	576	1.50	4.75	3.71	1.11
Innovativeness	576	1.75	4.50	3.70	0.83
Insecurity	576	1.75	4.50	3.63	0.87
Discomfort	576	1.50	4.50	3.57	0.96
TRI Total Score	576	1.69	4.44	3.65	0.85

Note: N = number of participants; TRI = Technology Readiness Index.

**Table 3 nursrep-15-00325-t003:** Pearson correlation between age and TRI total score.

		Age	TRI Total Score
Age	Pearson Correlation	1	−0.292 **
	Sig. (2-tailed)		<0.001
	N	576	576

Note: **. Correlation is significant at the 0.01 level (2-tailed). N = number of participants; TRI = Technology Readiness Index.

**Table 4 nursrep-15-00325-t004:** ANOVA table for TRI by demographic characteristic.

		Sum of Squares	df	Mean Square	F	Sig.
Education	Between Groups	289.27	3	96.42	397.78	<0.001
	Within Groups	138.65	572	0.24		
	Total	427.92	575			
Employment	Between Groups	38.45	4	9.61	14.09	<0.001
	Within Groups	389.46	571	0.68		
	Total	427.92	575			
Familiarity With Using Technology	Between Groups	171.61	4	42.90	95.57	<0.001
	Within Groups	256.31	571	0.44		
	Total	427.92	575			

Note: ANOVA = one-way analysis of variance; TRI = Technology Readiness Index; df = degrees of freedom.

## Data Availability

The data will be shared by the authors upon request.

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
