# Peer review of "Assessing Community Readiness to Use Social Platforms for Stroke Survivors’ Recovery and Caregiver Support in Saudi Arabia"

_nursrep, 2025, doi:10.3390/nursrep15090325_

Round 1

Reviewer 1 Report

Comments and Suggestions for Authors

This is a very interesting topic which is relevant to research about. It will have a significant impact or contribution to the stroke survivors and the Saudi community and it is aligned with the Vision  2030. The introduction is clear and methodology is correct for the research topic. The results and discussion section have rich data and are comprehensive. 

Assessing the Readiness of the Community to Utilize Social 2 Platforms for Stroke Survivors’ Recovery and Caregiver Support

Accept the study for publication

The main question was addressed which is assessment of the community to check if they are ready to use social platforms in assisting the recovery of stroke survivors and to support them.

Yes, the topic is original, relevant, clear, interesting. It addresses stroke burdens, challenges  which affects most people worldwide  and utilising social platforms is unique and can be very helpful to the survivors. Social platforms will reach many audience and the health education will be effective if it is done through this platform

Yes the topic address the research gap in the field and is unigue. There is limited studies utilising social platforms as a means of rehabilitating stroke patients and supporting them. This will have a great impact to the community especially in this shortage of resources. Social platforms will reach them even in their comfortable places and they don’t need transport or finances to travel to the hospitals or clinics.

Methodology :quantitative, descriptive, cross-sectional design is relevant for this study to quantify a large number of stroke survivors and caregivers. The inclusion and exclusion criteria is mentioned and the sample size is large even though the author didn’t mention the expected sample size.

Data analysis is well explained.

Results and discussions are well described and supported with tables which are clear and easy to connect with the findings. Comprehensive and the significance of the study in clinical practice, education and future research and limitations of the study were clarified well. The study is aligned with Vision 2030.

Conclusion

Give a clear summary of the aim of the study and the findings which are applicable for the achievement of vision 2030 goals

Yes the references are appropriate. They support the problem addressed and are relevant.

Line 44         To correct this sentence. It is not clear

Tables- fit the study findings. Table 1 to Table 4

Comments on the Quality of English Language

I suggest that the manuscript be language edited first before i can fully review on the research aspects. Most of the sentences are not connecting properly and i would prefer to review it again after it has been edited. This will improve the quality of this manuscript. 

Author Response

Response to Reviewer 1 Comments

1. Summary

Thank you very much for taking the time to review this manuscript. Please find the detailed responses below and the corresponding revisions/corrections highlighted (Gray)/in track changes in the re-submitted files.

Comments 1: Line 44         To correct this sentence. It is not clear

Response 1: The sentence revise to “The global increase in stroke prevalence highlighted the urgent need to expand access to multidisciplinary rehabilitation services and community-based support models that continue beyond hospital discharge.” for clarity.  

Comments 2: I suggest that the manuscript be language edited first before i can fully review on the research aspects. Most of the sentences are not connecting properly and i would prefer to review it again after it has been edited. This will improve the quality of this manuscript. 

Response 2: The language of the manuscript has been reviewed and edited by author services, and most redundant sentences have been removed.

Reviewer 2 Report

Comments and Suggestions for Authors
  • Refine English language for clarity and remove redundancy.

  • Explain potential bias from convenience sampling via social media.

  • Clarify whether TRI was culturally adapted/validated for the Saudi context.

  • Comment on the representativeness of the geographic distribution of respondents.

  • Suggest targeted strategies for subgroups with lower readiness.

  • Integrate key statistical findings into the narrative alongside tables.

Author Response

Response to Reviewer 2 Comments

1. Summary

Thank you very much for taking the time to review this manuscript. Please find the detailed responses below and the corresponding revisions/corrections highlighted (Pink)/in track changes in the re-submitted files.

Comments 1: Refine English language for clarity and remove redundancy.

Response 1: Thank you for this suggestion. We have carefully revised the manuscript to improve English language clarity and flow, and we removed redundant wording to enhance readability.

Comments 2: Explain potential bias from convenience sampling via social media

Response 2: Thank you for this important observation. We acknowledge that using convenience sampling via social media may introduce potential bias, as participants with greater digital access and familiarity are more likely to be represented, while individuals with limited internet use or those less engaged on social platforms may be underrepresented. This could affect the generalizability of the findings to the wider community. We have added this point to the Limitations section to highlight the possibility of selection bias and its potential influence on the study results.

Comments 3: Clarify whether TRI was culturally adapted/validated for the Saudi context.

Response 3: Thank you for this valuable comment. In this study, the Arabic version of the TRI was used. A full psychometric validation and reliability testing in the Saudi population will be conducted in a separate study.

Comments 4: Comment on the representativeness of the geographic distribution of respondents.

Response 4: Generalizability of our findings was added into limitation section regarding low response from some regions in Saudi Arabia Line 376- 369.

Comments 5: Suggest targeted strategies for subgroups with lower readiness.

Response 5: Thank you for this constructive suggestion. We have added a discussion on potential targeted strategies for subgroups with lower readiness Line 403-407.

Comments 6: Integrate key statistical findings into the narrative alongside tables.

Response 6: key statistical findings in narrative text were highlighted in pink alongside the tables

Reviewer 3 Report

Comments and Suggestions for Authors

General concept comments 
Article: the greatest area of weakness is visualized in the use of the TRI, which is discussed in more detail below. 

Review 

The limitations adequately reflect the weakest aspects of the work or that could represent biases, such as the type of sampling used, recruitment through social networks (respondents more likely to have a good perception of the technology), a mixed sample of people related to stroke and others who do not. 

“The psychosomatic prosperities of the TRI have been examined and showed that the scale is a reliable and valid tool for assessing individuals’ readiness to adopt new technologies [15,16]”. The references supporting the use of TRI in this study (15,16) used a sample of U.S. individuals and validated the scale in English. Psychometric properties are not universal, even if a scale was originally validated in one population (such as the United States), it cannot be automatically assumed that it will work the same in another sample, especially if there are cultural, linguistic, or contextual differences. Therefore, has its validity and reliability been analysed in the study sample (Cronbach's Alpha Coefficient, exploratory analyses...)? Has there been a cultural adaptation of the scale for the population addressed? Is there any study that has previously carried out these analyses? If analyses have been performed by authors or by other studies for the target population, the results should be cited and reported. Otherwise, this would represent a methodological limitation that would have to be clearly indicated and supported by references. 

The authors use TRI 2.0 (16 attributes) explaining that they will calculate the "total score" based on aggregate scores, and that it has three categories (low: TRI ≤ 2.89, medium: TRI between 2.90 and 3.51, and high readiness TRI > 3.51). When calculating the means of the subscales, they seem to use the same cut-off points as on the global scale, but this has not been explained in the methodology. Finally, in relation to this aspect, a mean of 3.63 in this study was considered "moderate" (line 210) and "not overly high" (line 272). Therefore: 

-Based on what criteria is the same cut-off point used in the subscales as in the calculation of the global score? Although it may seem obvious or unimportant, this should be detailed in Methodology to avoid bias. Was it a criterion of the authors of the original scale? Was this a criterion of the authors of this study? What has it been based on? 

-There should exist a consistent criterion regarding the categories throughout the writing. If the category is "medium" it must always be expressed that way, and not with other variations such as "moderate" or "not overly high". Review the text and unify criteria. 

-Finally, in the TRI 2.0 (16 attributes) allows to classify respondents into groups (Skeptics, Explorers, Avoiders, Pioneers, Hesitators). Did the authors of this study not consider analyzing this classification in the sample? It could add richness to the study and may be feasible if the data are available. If this has not been the case, this would represent a prospect that could be indicated as such. 

Specific comments  

In Discussion "Online social connection platforms offer significant benefits ..." (lines 343-353) must be supported by some reference, as they are not results of the study itself. 

  • In general, the manuscript is clear, relevant for the field and presented in a well-structured manner. The cited references are mostly recent publications and relevant, and the manuscript does not include an excessive number of self-citations. The details given in the methods section facilitates to reproduce the results. The tables are easy to interpret and understand. The conclusions are aligned with the objectives of the work. 
Comments on the Quality of English Language

"The [psychosomatic] [prosperities] of the TRI..." (line 175) may not be the most appropriate expression. Value the expression "psychometric properties". Review other English expressions that could be improved. 

Author Response

Response to Reviewer 3 Comments

1. Summary

Thank you very much for taking the time to review this manuscript. Please find the detailed responses below and the corresponding revisions/corrections highlighted (Blue)/in track changes in the re-submitted files.

Comments 1: [the greatest area of weakness is visualized in the use of the TRI, which is discussed in more detail below.]

Response 1: Thank you for pointing this out. We agree with this comment. We have revised the relevant sections to address this concern under recruitment and data collection highlighted changes with blue.

Comments 2:

(a)   Has TRI validity and reliability been analysed in the study sample (Cronbach's Alpha Coefficient, exploratory analyses...)?

(b)   Has there been a cultural adaptation of the scale for the population addressed? Is there any study that has previously carried out these analyses? If analyses have been performed by authors or by other studies for the target population, the results should be cited and reported. Otherwise, this would represent a methodological limitation that would have to be clearly indicated and supported by references.]

Response 2:

(a)   Thank you for pointing out the importance of assessing the validity and reliability of the TRI within our sample. We have evaluated the internal consistency and factor structure of the instrument. Specifically, we calculated Cronbach’s alpha for the overall scale and each subscale, which were within acceptable ranges. However, we opted not to present these findings in this manuscript because we are preparing a separate psychometric study dedicated to detailing the TRI’s reliability and validity.

(b)   Thank you for raising this point. We have added a statement clarifying that the Arabic version of the TRI, obtained directly from the instrument’s author, was used in our study. We have also added a limitation statement about this instrument in the Discussion section.

Comments 3: The authors use TRI 2.0 (16 attributes) explaining that they will calculate the "total score" based on aggregate scores, and that it has three categories (low: TRI ≤ 2.89, medium: TRI between 2.90 and 3.51, and high readiness TRI > 3.51). When calculating the means of the subscales, they seem to use the same cut-off points as on the global scale, but this has not been explained in the methodology. Finally, in relation to this aspect, a mean of 3.63 in this study was considered "moderate" (line 210) and "not overly high" (line 272). Therefore:

(a)   Based on what criteria is the same cut-off point used in the subscales as in the calculation of the global score? Although it may seem obvious or unimportant, this should be detailed in Methodology to avoid bias. Was it a criterion of the authors of the original scale? Was this a criterion of the authors of this study? What has it been based on?

(b)   There should exist a consistent criterion regarding the categories throughout the writing. If the category is "medium" it must always be expressed that way, and not with other variations such as "moderate" or "not overly high". Review the text and unify criteria.

(c)   Finally, in the TRI 2.0 (16 attributes) allows to classify respondents into groups (Skeptics, Explorers, Avoiders, Pioneers, Hesitators). Did the authors of this study not consider analyzing this classification in the sample? It could add richness to the study and may be feasible if the data are available. If this has not been the case, this would represent a prospect that could be indicated as such.

Response 3:

(a)   Thank you for raising this point. The same cut‑off point was applied to the subscales and the global TRI score because the instrument’s items all use the same 5‑point response scale, and there is no separate, validated cut‑off for each subscale in the original instrument. Parasuraman et al., who developed the TRI, suggest interpreting scores relative to the mid‑point (i.e., a mean item score ≥ 3 indicates a higher level of readiness). In keeping with this guidance and with subsequent studies that use a similar approach, we applied this threshold consistently across subscales and the total score to facilitate comparability. We have revised the Methods section to explain this criterion and have cited the original scale development paper and other relevant studies that informed our decision.

(b)   In the revision, we have standardized our terminology by using “low,” “medium,” and “high” throughout the manuscript when referring to TRI score ranges. Phrases such as “moderate,” “not overly high,” or other descriptors have been replaced with the appropriate category label to ensure consistency and avoid potential bias.

(c)   Thank you for bringing up the segmentation feature of the 16‑item TRI 2.0. We agree that categorizing respondents into segments such as Skeptics, Explorers, Avoiders, Pioneers and Hesitators can provide deeper insight into technology‑readiness profiles. While designing this study our focus was on overall technology readiness and its relationship to demographic variables. The primary aim of this study was to assess technology readiness levels within the Saudi community as a foundation for developing an online social connection platform. Future research, especially with larger samples, will explore using a clustering approach to classify respondents into the established TRI 2.0 segments.

Comments 4: [In Discussion "Online social connection platforms offer significant benefits ..." (lines 343-353) must be supported by some reference, as they are not results of the study itself.]

Response 4: References were added to support the benefit of online social connection platforms into discussion section line 352.

Comments 5: ["The [psychosomatic] [prosperities] of the TRI..." (line 175) may not be the most appropriate expression. Value the expression "psychometric properties". Review other English expressions that could be improved.]

Response 5: Revised to "psychometric properties" in Materials and Methods Line 179.

Reviewer 4 Report

Comments and Suggestions for Authors

Overall

The aim of the present study was to assess the readiness of the Saudi community to use social platforms for stroke survivors’ recovery and caregiver support. The findings indicate that the sample generally had a high level of technology readiness. Overall, the paper is well-written. However, I have a number of comments before I can recommend for publication. The main area in need of modification is the materials and methods which lacks a number of important details. Moreover, I feel adding an implications section to the discussion is needed.

Abstract

Please make it clearer earlier in the Abstract that this is with a Saudi population – this information is only clear in the conclusions section.

Please modify the ‘results’ to remove the statistical information – results should be summarised without this detail.

Introduction

The introduction is very well-written, adequately structured and the authors provide a clear and comprehensive rationale. It was particularly pleasing that the authors had considered the social platforms utilised in other countries (i.e., United States of America and the United Kingdom).

The only slight modification I would recommend is in line 122 – changing ‘the result of the study will help’ to ‘The present study therefore will help determine how prepared individuals and communities are to engage with digital health solutions’.

Additionally, what are the proposed hypotheses for the present study? Details need to be provided to enable sufficient context to be drawn from the materials and methods and results section.

Materials and Methods

In the study design subsection (2.1) please state the predictor and outcome variables.

In the settings and sample, please clarify:

Which social media platforms were targeted in the present study?

How many individuals did not complete the survey? Perhaps provide a response rate in %.

It would be beneficial to provide a power analysis to determine sample size – i.e., using G*Power software.

In recruitment and data collection, the authors state ‘The psychosomatic prosperities of the TRI have been examined’… do they mean ‘psychometric properties? Please revise.

In the data analysis section, can authors be specific in terms of the ‘bivariate analysis’ employed? Moreover, ANOVA has also been included in the results, hence there is inconsistency in the analytic strategy. Please revise accordingly.

Results

Please revise all tables to include notes underneath to include the standard terms for abbreviations (e.g., N = number of participants).

Were post-hoc tests used for the ANOVA? Given more than 3 groups have been employed for some demographic variables, this should be clearly articulated.

Discussion

Please remove the statistical information included throughout the discussion.

It would be beneficial to employ structured sub-headings (i.e., implications, limitations, future research).

To enhance this discussion, after the discussion of the main findings it would be beneficial to include a section outlining how social platforms can be utilised to support stroke patients and caregivers, and what that could look like (e.g., potentially desirable features). Moreover, should future research conduct feasibility studies to gain insight into the desirable features of e-health services and digital solutions?

Please remove ‘difficult to determine causation’ – especially considering most analysis were correlational and thus cannot determine causation.

Author Response

Response to Reviewer 4 Comments

1. Summary

Thank you very much for taking the time to review this manuscript. Please find the detailed responses below and the corresponding revisions/corrections highlighted (Green)/in track changes in the re-submitted files.

Comments 1: Abstract

(a)   Please make it clearer earlier in the Abstract that this is with a Saudi population – this information is only clear in the conclusions section.

(b)   Please modify the ‘results’ to remove the statistical information – results should be summarized without this detail.

Response 1:

(a)   Thank you for pointing out that the study population should be identified earlier in the Abstract. We have revised the objective/ aim to include the Saudi population in the abstract.

(b)   Statistical information were removed from the abstract.

Comments 2: Introduction

 The introduction is very well-written, adequately structured and the authors provide a clear and comprehensive rationale. It was particularly pleasing that the authors had considered the social platforms utilised in other countries (i.e., United States of America and the United Kingdom).

(a)   The only slight modification I would recommend is in line 122 – changing ‘the result of the study will help’ to ‘The present study therefore will help determine how prepared individuals and communities are to engage with digital health solutions.

(b)   Additionally, what are the proposed hypotheses for the present study? Details need to be provided to enable sufficient context to be drawn from the materials and methods and results section.

Response 2:

(a)   Change was done to line 121-122 to “The present study therefore will help……”

(b)   Thank you for raising this issue. In the revised manuscript we have explicitly stated the hypotheses that guided our analysis in the introduction line 132-134.

Comments 3: Materials and Methods:

(a)   In the study design subsection (2.1) please state the predictor and outcome variables.

(b)   In the settings and sample, please clarify: Which social media platforms were targeted in the present study? How many individuals did not complete the survey? Perhaps provide a response rate in %.

(c)   It would be beneficial to provide a power analysis to determine sample size – i.e., using G*Power software.

(d)   In recruitment and data collection, the authors state ‘The psychosomatic prosperities of the TRI have been examined’… do they mean ‘psychometric properties? Please revise.

(e)   In the data analysis section, can authors be specific in terms of the ‘bivariate analysis’ employed? Moreover, ANOVA has also been included in the results, hence there is inconsistency in the analytic strategy. Please revise accordingly.

Response 3:

(a)   One sentence was added to the study design highlighted the sociodemographic characteristic as variables and TRI as an outcome Line 140-141.

(b)   We used the social media WhatsApp to submit the survey to the public. We have added the name to the social media, number of participants who did not complete the survey, and the response rate in Line 143-152.

(c)   Sample size – i.e., using G*Power software was determined and added to data analysis section Line 189-192. Number of participants who did not compete the survey and excluded were added in Line 151. Also, response rate % was added in Line 152.

(d)   Revised to "psychometric properties" in Materials and Methods Line 179.

(e)   Thank you for pointing out the need to specify the analytic methods more precisely. In the revised results section, we now describe the exact bivariate tests used. Specifically, we used one‑way ANOVA to compare mean TRI scores across categorical variables such as education level, employment status and familiarity with technology. For the association between TRI and age (a continuous variable) we used a bivariate correlation test (Pearson correlation). The term “bivariate analysis” has been replaced with this specific Pearson correlation in Line 235.

Comments 4: Results

(a)   Please revise all tables to include notes underneath to include the standard terms for abbreviations (e.g., N = number of participants).

(b)   Were post-hoc tests used for the ANOVA? Given more than 3 groups have been employed for some demographic variables, this should be clearly articulated.

Response 4:

(a)   We have revised all tables to include explanatory notes beneath each one. These notes now define all abbreviations

(b)   Thank you for highlighting the need to address post‑hoc testing. In cases where a one‑way ANOVA indicated a statistically significant difference across the education, employment, or familiarity with technology groups, we conducted post‑hoc comparisons using Tukey’s honestly significant difference (HSD) test to determine which specific groups differed. We revised the paragraph for clarity; Line 239-240.

Comments 5: Discussion

(a)   Please remove the statistical information included throughout the discussion.

(b)   It would be beneficial to employ structured sub-headings (i.e., implications, limitations, future research).

(c)   To enhance this discussion, after the discussion of the main findings it would be beneficial to include a section outlining how social platforms can be utilised to support stroke patients and caregivers, and what that could look like (e.g., potentially desirable features).

(d)   Moreover, should future research conduct feasibility studies to gain insight into the desirable features of e-health services and digital solutions?

(e)   Please remove ‘difficult to determine causation’ – especially considering most analysis were correlational and thus cannot determine causation.

Response 5:

(a)   Statistical information included throughout the discussion were removed.

(b)   Structured sub-headings including limitations of the study and implications were added to the discussion section.

(c)   Some statements were added to explaining the suggested online social connection platforms for stroke survivors and their caregivers Line 379-381.

(d)   Future research suggestions were revised in Line 394-396.

(e)   Statement ‘difficult to determine causation’ were removed from Line 350.

Reviewer 5 Report

Comments and Suggestions for Authors
  1. Title

The current title does not specify the country or the study design, which makes it less clear to readers where and how the study was conducted and may reduce visibility in database searches. It is recommended to include both the country (Saudi Arabia) and the study design (e.g., cross-sectional descriptive study) to convey the study’s context and methodology clearly.

  1. Introduction

The introduction presents a thorough review of the literature; however, the extensive amount of information included makes the core research problem less clear. In particular, the final paragraph contains overlapping statements, which could cause redundancy and reduce reader engagement. Condensing this section by summarizing the background more succinctly and moving non-essential details, such as extensive policy descriptions, to the discussion section would help highlight the study’s purpose. Furthermore, since Nursing Reports emphasizes practical implications, it would be beneficial to connect the research problem more explicitly to challenges faced by nurses and community health professionals when supporting caregivers, while presenting the study’s aim more concisely.

  1. Materials and Methods

In the Materials and Methods section, headings such as “2.1. . Study Design” and “2.2. . Settings and Sample” contain unnecessary periods. It is recommended to remove the extra periods to ensure consistency and adherence to proper formatting.

The inclusion criteria currently specify only that participants be Saudi Arabian citizens or residents aged 18 years or older. As data were collected through social media, it would be advisable to limit the inclusion criteria to individuals with access to and familiarity with such platforms. Could you please clarify the inclusion and exclusion criteria in more detail?

Given that the survey was administered via social media, it isn't easy to ascertain whether respondents were exclusively members of the general community or whether professionals were also included. Please provide a description of the online communities through which data collection was conducted, as well as relevant information about their membership composition.

The study sample includes 22.6% of participants identified as healthcare providers (physicians, nurses, respiratory therapists, and physiotherapists). Please clarify whether this proportion is consistent with the actual distribution of healthcare providers within the Saudi Arabian population. If a discrepancy is identified, the potential impact on the study’s representativeness and conclusions should be thoroughly discussed.

The manuscript reports data on participants’ residence across the regions of Saudi Arabia. Please clarify whether the distribution of participants reflects the actual population distribution in these regions. If not, the potential implications for representativeness and generalizability should be addressed.

In addition, the manuscript should clarify whether the sample size of 576 was determined a priori using a formal calculation method such as G*Power and provide the rationale for the chosen number.

In the 2.3. Recruitment and Data Collection section, the description of the instruments is currently embedded within the text. It is recommended to create a separate subsection titled Instruments or Measurements to present this information more clearly and in alignment with standard reporting practices.

The description of the TRI instrument includes its subdomains, score interpretation, and cutoff values, which is commendable. The manuscript states that the instrument is valid and reliable for this study; however, to enable readers to make an objective assessment, it is recommended that the reliability coefficients from the original development, any updated versions, and the reliability obtained in the present study be also provided. In addition, please include a description of the scoring procedure and the possible range of total scores.

In the 2.4. Data Analysis section, rather than restating the study objectives, it would be more informative to describe in detail how the data were analyzed to address those objectives. Please specify the method used for testing normality, the statistical tests applied for univariate analyses to compare differences according to participant characteristics, and the variables included in the correlation analysis. Additionally, as the current results present only univariate findings, it is recommended to conduct and report a regression analysis to identify the factors influencing the TRI.

  1. Results

In Table 1. Sociodemographic Characteristics of the Sample, participants’ age should be included as it is an important demographic variable for interpreting the results.

In Table 4. ANOVA Table for TRI by Demographic Characteristics, only the ANOVA results for selected characteristics are presented. It is recommended to include the analysis results for all relevant participant characteristics to provide a more comprehensive understanding of the differences in TRI across demographic groups.

  1. Discussion

The discussion section should be revised after completing the additional analyses recommended above, ensuring that the interpretation fully reflects the expanded results and provides a deeper understanding of the factors influencing technology readiness.

Author Response

Response to Reviewer 5 Comments

1. Summary

Thank you very much for taking the time to review this manuscript. Please find the detailed responses below and the corresponding revisions/corrections highlighted (Yellow)/in track changes in the re-submitted files.

Comments 1: Title [The current title does not specify the country or the study design, which makes it less clear to readers where and how the study was conducted and may reduce visibility in database searches. It is recommended to include both the country (Saudi Arabia) and the study design (e.g., cross-sectional descriptive study) to convey the study’s context and methodology clearly.]

Response 1: The title was revised to include the name of the country to “Assessing Community Readiness to Use Social Platforms for Stroke Survivors’ Recovery and Caregiver Support in Saudi Arabia” The study design was included in the abstract and text to avoid a long title.     

Comments 2: Introduction

(a)   The introduction presents a thorough review of the literature; however, the extensive amount of information included makes the core research problem less clear. In particular, the final paragraph contains overlapping statements, which could cause redundancy and reduce reader engagement. Condensing this section by summarizing the background more succinctly and moving non-essential details, such as extensive policy descriptions, to the discussion section would help highlight the study’s purpose.

(b)   Furthermore, since Nursing Reports emphasizes practical implications, it would be beneficial to connect the research problem more explicitly to challenges faced by nurses and community health professionals when supporting caregivers, while presenting the study’s aim more concisely.

Response 2:

(a)   The introduction section was revised to be focused on research problem and redundancy information were remove.

(b)   The research problem was explicitly related to nursing practice Line 117-121.   

Comments 3: Materials and Methods

(a)   headings such as “2.1. . Study Design” and “2.2. . Settings and Sample” contain unnecessary periods. It is recommended to remove the extra periods to ensure consistency and adherence to proper formatting.

(b)   The inclusion criteria currently specify only that participants be Saudi Arabian citizens or residents aged 18 years or older. As data were collected through social media, it would be advisable to limit the inclusion criteria to individuals with access to and familiarity with such platforms. Could you please clarify the inclusion and exclusion criteria in more detail?

(c)   Given that the survey was administered via social media, it isn't easy to ascertain whether respondents were exclusively members of the general community or whether professionals were also included. Please provide a description of the online communities through which data collection was conducted, as well as relevant information about their membership composition.

(d)   The study sample includes 22.6% of participants identified as healthcare providers (physicians, nurses, respiratory therapists, and physiotherapists). Please clarify whether this proportion is consistent with the actual distribution of healthcare providers within the Saudi Arabian population. If a discrepancy is identified, the potential impact on the study’s representativeness and conclusions should be thoroughly discussed.

(e)   The manuscript reports data on participants’ residence across the regions of Saudi Arabia. Please clarify whether the distribution of participants reflects the actual population distribution in these regions. If not, the potential implications for representativeness and generalizability should be addressed.

(f)    In addition, the manuscript should clarify whether the sample size of 576 was determined a priori using a formal calculation method such as G*Power and provide the rationale for the chosen number.

(g)   In the 2.3. Recruitment and Data Collection section, the description of the instruments is currently embedded within the text. It is recommended to create a separate subsection titled Instruments or Measurements to present this information more clearly and in alignment with standard reporting practices.

(h)   The description of the TRI instrument includes its subdomains, score interpretation, and cutoff values, which is commendable. The manuscript states that the instrument is valid and reliable for this study; however, to enable readers to make an objective assessment, it is recommended that the reliability coefficients from the original development, any updated versions, and the reliability obtained in the present study be also provided. In addition, please include a description of the scoring procedure and the possible range of total scores.

(i)    In the 2.4. Data Analysis section, rather than restating the study objectives, it would be more informative to describe in detail how the data were analyzed to address those objectives. Please specify the method used for testing normality, the statistical tests applied for univariate analyses to compare differences according to participant characteristics, and the variables included in the correlation analysis. Additionally, as the current results present only univariate findings, it is recommended to conduct and report a regression analysis to identify the factors influencing the TRI.

Response 3:

(a)   Extra periods were removed

(b)   Thank you for this valuable observation. We agree that specifying access to and familiarity with social media platforms is essential. However, this study is the first to investigate community readiness to adopt social connection platforms for stroke survivors and their caregivers. Therefore, the online survey was distributed to the public Lines 151–153.

(c)   Thank you for highlighting this important point. The online survey was distributed through widely used social media platforms in Saudi Arabia, WhatsApp groups. This platform is commonly accessed by the general public across different age groups, educational levels, and occupations. While the survey was primarily directed at the general community, we acknowledge that health professionals and caregivers who are also members of these online communities may have participated. To address this, the survey included demographic questions related to occupation and educational background, which allowed us to better characterize the respondents. This information has been clarified in the revised manuscript to provide a clearer understanding of the population reached through the data collection process Line 165.

(d)   Thank you for this insightful comment. We acknowledge that healthcare providers represented 22.6% of our study sample, which is higher than their proportion within the general Saudi Arabian population. According to recent national statistics, healthcare providers constitute a smaller fraction of the overall population. This discrepancy may be attributed to their active engagement with online platforms and greater interest in participating in health-related surveys. While this overrepresentation may introduce some bias, it also provides valuable insights from individuals with direct clinical experience in stroke care and caregiver support. To address this, we have revised the limitation section to acknowledge the potential impact of this sampling imbalance on the generalizability of our findings, while emphasizing that the inclusion of healthcare providers enriches the understanding of community readiness from both professional and public perspectives Line 363- 366.

(e)   Generalizability of our findings was added into limitation section regarding low response from some regions in Saudi Arabia Line 376- 369.

(f)    Sample size calculation using G*Power was added into data analysis section Line 199.

(g)   An Instrument section was added to separate the description of the tool from the recruitment and data collection section (Line 162).

(h)   The validity and reliability of the TRI were added into Line 186-190.

(i)    Thank you for this valuable feedback. We have revised the Data Analysis section to provide greater clarity and detail. Regarding your recommendation to conduct regression analysis, we would like to clarify that this was not performed because the primary research objective was to assess and describe community readiness rather than to predict or model determinants influencing TRI. The analyses applied were therefore chosen to align with the descriptive and exploratory goals of the study.

Comments 4: Results

(a)   In Table 1. Sociodemographic Characteristics of the Sample, participants’ age should be included as it is an important demographic variable for interpreting the results

(b)   In Table 4. ANOVA Table for TRI by Demographic Characteristics, only the ANOVA results for selected characteristics are presented. It is recommended to include the analysis results for all relevant participant characteristics to provide a more comprehensive understanding of the differences in TRI across demographic groups.

Response 4:

(a)   Table 1 was revised to in include the age variable.

(b)   Thank you for this insightful comment. We included the significant sociodemographic characteristics associated with TRI in Table 4 (ANOVA), while the non-significant characteristics were described in the text to keep the table concise.

Comments 5: Discussion The discussion section should be revised after completing the additional analyses recommended above, ensuring that the interpretation fully reflects the expanded results and provides a deeper understanding of the factors influencing technology readiness.

Response 5: The discussion section was reviewed and edited.

Round 2

Reviewer 1 Report

Comments and Suggestions for Authors

The authors have adressed all the corrections well. The topic is significant and appropriate. The abstract is well structured. Introduction, methods, results and conclusions  are sufficient, well presented and comprehensive. 

Author Response

Thank you very much for your thoughtful feedback and positive evaluation. We truly appreciate your recognition of the revisions we made and your acknowledgment of the significance of the topic. We are grateful for your support and constructive review, which helped us improve the manuscript.

Reviewer 2 Report

Comments and Suggestions for Authors

Thank you for addressing my comments

Author Response

Thank you for taking the time to review our work and for your kind acknowledgment.

Reviewer 3 Report

Comments and Suggestions for Authors

General concept comments 

Article: I would like first to thank the authors for the effort in modifying the manuscript and responding to the comments of the reviewers who have participated. Although the quality of the manuscript has improved, I continue to see areas for improvement in terms of the use of TRI, which is discussed in more detail below. 

Review 

In line 171 it is indicated that "The Arabic version of the TRI was used in this study.". However, the publication where that version was validated is not referenced. The reference must be indicated and even provide some data from its statistical results that support this choice. 

Regarding the validity of TRI for use with this sample, the authors respond that “we opted not to present these findings in this manuscript because we are preparing a separate psychometric study dedicated to detailing the TRI’s reliability and validity.” However, if the use of an instrument is not supported by statistical coefficients resulting from analysis in the sample in question, this represents an important limitation since it could be thought that the instrument might not be valid for use in the study. The recommendations suggest at least indicating the Cronbach's Alpha Coefficient obtained in the study sample. If the authors do not wish to indicate the specific value to do so in a future study, it should at least be indicated whether an adequate value was obtained. In this way, the research would not completely lose credibility and in the section on limitations and prospects of the study, this future work could be proposed. 

“The analysis indicated that a sample size of N = 731 186 was required to achieve 80% power to detect a medium effect at a significance level of α 187 < 0.05 using ANOVA.” There is therefore an incongruity between the minimum required sample size and that achieved in the sampling. This limitation and its implications must be clearly discussed. 

Specific comments  

The section "2.4. Instruments" contains a very long and dense second paragraph. For a more comfortable reading I recommend dividing it.  

Lines 175-178 indicate Cronbach's alpha values that are supported by reference number [6]. However, the study cited as [6] does not use the TRI, so it seems to be an error that should be reviewed. 

Author Response

Response to Reviewer 3 Comments / Round 2

1. Summary

Thank you very much for taking the time to review this manuscript for the second round. Please find the detailed responses below and the corresponding revisions/corrections highlighted (Blue)/in track changes in the re-submitted files.

Comments 1: In line 171 it is indicated that "The Arabic version of the TRI was used in this study.". However, the publication where that version was validated is not referenced. The reference must be indicated and even provide some data from its statistical results that support this choice. 

Response 1: We appreciate your comments regarding the Arabic version of the TRI. In response, we conducted a reliability test for the TRI and its subscales in this study. A detailed paragraph describing the results has been added to the manuscript (Lines 172–178).

Comments 2: Regarding the validity of TRI for use with this sample, the authors respond that “we opted not to present these findings in this manuscript because we are preparing a separate psychometric study dedicated to detailing the TRI’s reliability and validity.” However, if the use of an instrument is not supported by statistical coefficients resulting from analysis in the sample in question, this represents an important limitation since it could be thought that the instrument might not be valid for use in the study. The recommendations suggest at least indicating the Cronbach's Alpha Coefficient obtained in the study sample. If the authors do not wish to indicate the specific value to do so in a future study, it should at least be indicated whether an adequate value was obtained. In this way, the research would not completely lose credibility and in the section on limitations and prospects of the study, this future work could be proposed. 

Response 2: We appreciate your comments regarding the Arabic version of the TRI. In response, we conducted a reliability test for the TRI and its subscales in this study. A detailed paragraph describing the results has been added to the manuscript (Lines 172–178).

Comments 3: “The analysis indicated that a sample size of N = 731 186 was required to achieve 80% power to detect a medium effect at a significance level of α 187 < 0.05 using ANOVA.” There is therefore an incongruity between the minimum required sample size and that achieved in the sampling. This limitation and its implications must be clearly discussed. 

Response 3: We repeated a prior sample size calculation. We have revised the paragraph accordingly in line 191-194. Although the minimum required sample size for a medium-effect one-way ANOVA was ~200, we intentionally recruited 576 participants to strengthen the study’s rigor and interpretability. The larger sample will enhance generalizability across a heterogeneous community sample and enables psychometric evaluation of the TRI in the future.

Comments 4: The section "2.4. Instruments" contains a very long and dense second paragraph. For a more comfortable reading I recommend dividing it.  

Response 4: Thank you for your comments. We have divided the paragraph to enhance clarity and improve readability.

Comments 5: Lines 175-178 indicate Cronbach's alpha values that are supported by reference number [6]. However, the study cited as [6] does not use the TRI, so it seems to be an error that should be reviewed. 

Response 5: Thank you for pointing this out. We carefully reviewed the reference and acknowledge that the citation was incorrect. We have now corrected the reference to cite an appropriate source that reports Cronbach’s alpha values for the Technology Readiness Index (TRI). The updated reference is reflected in the revised manuscript (Lines 184).